# Family Factors Related to Suicidal Behavior in Adolescents

**DOI:** 10.3390/ijerph19169892

**Published:** 2022-08-11

**Authors:** Xavier Alvarez-Subiela, Carmina Castellano-Tejedor, Francisco Villar-Cabeza, Mar Vila-Grifoll, Diego Palao-Vidal

**Affiliations:** 1Suicide Conduct Unit, Psychiatry and Psychology Department, Sant Joan de Déu Hospital, 08950 Esplugues del Llobregat, Spain; 2Doctoral Program in Psychiatry, Department of Psychiatry and Forensic Medicine, Autonomous University of Barcelona, Bellaterra, 08193 Barcelona, Spain; 3Research Group on Stress and Health (GIES), Department of Basic Psychology, Faculty of Psychology, Autonomous University of Barcelona, Bellaterra, 08193 Barcelona, Spain; 4RE-FIT Research Group, Parc Sanitari Pere Virgili & Vall d’Hebron Institute of Research, 08023 Barcelona, Spain; 5Centro de Investigación en Red de Salud Mental (CIBERSAM), Carlos III Health Institute, 28029 Madrid, Spain; 6Unitat Mixta de Neurociència Traslacional I3PT-INc-UAB, Institut d’Investigació i Innovació Parc Taulí (I3PT), Sabadell, 08208 Barcelona, Spain; 7Department of Mental Health, University Hospital Parc Taulí, Sabadell, 08208 Barcelona, Spain

**Keywords:** suicidal behavior, adolescent, family factors, stressful life events, suicide prevention

## Abstract

Objective: This research aims to investigate what type of family patterns (specifically attachment, bonding and family functioning) and stressful life events can trigger or protect adolescents from developing suicidal behavior. Methods: For these purposes, a case-control study (adolescents with suicidal behavior vs. paired adolescents with no suicidal behavior) was conducted with one hundred 12 to 17-year-old adolescents (50 controls, 50 cases, 74% females), assessed between 2018 and 2020. Results: Negligent (*p* < 0.001) or affection-less control bonding (*p* < 0.001), insecure attachment (*p* = 0.001) and stressful life events (*p* < 0.001) revealed to be significant risk factors for suicidal behavior. On the contrary, parents’ care (*p* < 0.001) and security (*p* < 0.001) were revealed as protective factors for suicidal behavior. Conclusions: Considering these results, family interventions and improving coping skills seem to be two essential targets for any suicide prevention intervention in adolescents.

## 1. Introduction

Worldwide, suicide was the fourth cause of death among 15 to 29-year-olds in 2019, after road injury, tuberculosis, and interpersonal violence [1]. Empirical evidence establishes a few commonly accepted risk factors for suicidal behavior in this age group that can be divided into (1) social factors (i.e., suicidal behavior in the community, media influence, availability of health resources and professional support); (2) individual factors (i.e., psycho-emotional problems, transition and challenges in puberty, religious beliefs, stress management strategies, affective relationships, education); and (3) family factors (i.e., family relationships and bounds, socio-economic status, security, and health conditions) [2].

A significant corpus of research exists on individual factors [3,4,5]. However, more empirical evidence regarding family factors is needed to better understand how these variables might modulate suicidal behavior in adolescents, since family is the immediate social environment in which adolescents grow up. So far, several studies have shown evidence of an increase in the number of suicide attempts in adolescents that have a family history of suicide attempts or completed suicide [6,7]. Additionally, having a low socioeconomic status and having dysfunctional family patterns (e.g., having serious conflicts with the family, or serious family mental health problems) have been pointed out as risk factors for attempting suicide [8,9,10]. Despite this preliminary evidence, it is still necessary to further study other family-related factors such as attachment style, parental bonding, and family functioning patterns that might influence the risk of suicidal behavior among adolescents.

The early parent–child relationship has been associated with numerous aspects of behavior and development. An estimation of these early relationships in adults can be very useful; this may inform diagnostic and therapeutic interventions, whereas in psychological research, it can contribute to a more precise understanding of the role of the early environment in the etiology of psychopathology [11]. Empirical evidence shows that parental bonding has an important role in developing psychopathology during adolescence and young adulthood [12,13]. It has been shown that low parental concern or less care are related to a greater number of suicide attempts in adolescents with mental health problems [14], showing the importance of positive parenting. Authoritarian and neglect/rejection profiles have also been related to an increase in suicide attempts [15], and evidence of the Affectionless Control model’s influence on adolescents’ suicidal behavior is increasing [16,17,18]. However, more studies are needed to clarify these influences.

Attachment theory postulates that early care experiences are internalized as inner functional models that guide individuals at the relational level [19]. That is to say, attachment styles are always the result of the interaction between the person’s innate attachment style and the history of experiences with parents and caregivers in relation to attachment [20]. The first classification of attachment styles distinguished between secure, insecure-ambivalent, and insecure-avoidant attachment [21]. Secure attachment is produced when the caretakers demonstrate physical and emotional warmth, trust, and availability. It has been described how children with insecure-avoidant attachment develop self-sufficiency and a preference for emotional distancing from others [22].

Several researchers have established a direct relationship between insecure attachment and a higher risk for suicidal behavior among adolescents [23,24]. Main explanations for this association have pointed out the additive effects of higher self-criticism and dependency among adolescents showing this specific family pattern. However, specific mechanisms of such relationships are still unclear [25]. Other research has found no direct relationship between attachment style and suicidal behavior [26], suggesting that the relationship between these two factors is mediated by other variables, like social information processing that can mediate in personality traits as behavioral patterns and impulsivity [27,28]. In this sense, Falgares and colleagues also found that self-criticism and dependence mediated between an anxious/ambivalent attachment style and suicidal behavior; however, in the denial/avoidance attachment style, only self-criticism was found to be a mediating factor [29]. On the other hand, Zisk and colleagues found that, concerning attachment, negative expectations from caregivers mediated suicidal behavior and insecurity [30]. Similarly, having a family background of mental health problems has been systematically related to a higher risk of suicide in adolescents. Still, again, specific mechanisms leading to these behaviors have proved difficult to study [31]. 

Although there is some consensus regarding the relationship between insecure attachment and suicidal behavior in the adolescent population, there is little knowledge about the association between them. By identifying these processes, relevant conclusions could be obtained to improve treatment options, including treatment planning and family work, to help patients with high suicide risk [32,33,34,35].

Other family models are based on the Circumplex Model of Family System, which relates to the family’s cohesion and adaptability [36,37]. This model proposes that a balanced level of both cohesion and adaptability is the most functional family development [37]. Some studies relate these unbalanced family functional models to mental health problems [38,39,40,41], but few have provided empirical data related to suicidal behavior [42]. It is important to further study the relationship between family function and suicidal behavior in adolescents to improve family interventions and suicide prevention.

Finally, it is also imperative to consider research on stressful life events (SLE). SLE increase suicidal tendencies in adolescents due to increased psychological distress and fatigue from the social support they receive, requiring an emphasis on support programs and stress management as a method of preventing suicide, and more studies to determine which SLE have more risk are necessary [43]. Some studies show three SLE categories as being more frequent in adolescents with suicidal behavior: family conflicts, academic stressors, and trauma. Related to family conflicts, the most important are stress related to parents, lack of adult support outside of the home, physical harm by a parent, running away from home, living apart from both parents, and other family situations associated with risk for suicidality (i.e., parental suicidal behavior, early death, mental illness in a relative, unemployment, low income, neglect, parental divorce, other parent loss, and family violence). As an important life event in adolescence, academic stressors also mediate suicide behavior [44]. Trauma, bullying, and childhood sexual abuse are the most important SLE. Still, there are also other very commonly described stressful circumstances that may precede suicidal behavior, such as peer conflict, legal problems, physical abuse, worries about sexual orientation, romantic breakup, and exposure to suicide or suicide attempts [3]. 

Considering all this evidence, the main goal of this research is to investigate what type of family patterns—specifically, attachment, bonding, and family functioning—and stressful life events—specifically, bullying, cyberbullying, and others—can trigger or protect adolescents from developing suicidal behavior. We hypothesized that negligent and authoritarian parenting, insecure attachment, decompensated family functioning, and stressful life events are risk factors for suicide behaviors in adolescents, while optimal bonding, secure attachment, and compensated family functioning are protective factors for suicide behavior in adolescents. To investigate this further would help to design more accurate preventive interventions considering families and main caregivers of this sample population.

## 2. Methods

### 2.1. Participants

This is a case-control study of one hundred 12 to 17-year-old adolescents (50 controls and 50 cases) recruited over a period of two years (2018–2020). The clinical sample was recruited from a Psychiatric Inpatient Unit of the Hospital Sant Joan de Déu after patients were admitted because of suicidal behavior (this includes: suicide ideation, suicide planning, or suicide attempt) [45]. The clinical sample was compared to a convenience sample of controls recruited from schools and recreational associations of the same area of influence of our hospital, to reduce bias and to homogenize the sample in terms of socio-economic and environmental characteristics.

Inclusion criteria for the clinical sample were: (1) patients from 12 to 17 years of age, (2) patients admitted to a psychiatric inpatient unit due to suicidal behavior, (3) agreeing to participate in the study. The exclusion criteria were (1) subjects of legal age (≥18 years) or under 12 years of age, (2) patients with cognitive or other neuropsychological deficits that could hinder the clinical assessment and/or the understanding of the concept of death, (3) subjects denying suicide intentionality for the behavior (self-harm, intoxication, or other similar behaviors with anxiolytic, playful, or other non-suicidal intention), and (4) not living with the family or being institutionalized at the time of the study.

Inclusion criteria for the control sample were: (1) adolescents from 12 to 17 years of age, both included, (2) living in the same area of influence of the clinical sample, and (3) agreeing to participate in the study. The exclusion criteria were: (1) adolescents not living with the family or being institutionalized at the time of the study and (2) families who have other children with present or past suicidal behavior. 

### 2.2. Instruments

Data collection was gathered by a mental health professional (psychologist or psychiatrist) through a semi-structured interview and the administration of different questionnaires (self-administered) providing general information and instructions on how to fulfill them. All cases and their parents were assessed in the inpatient unit (same room, but independent assessments). In the case of the control sample and their parents, the assessments were carried out independently in meeting rooms of different associations, or they were given the questionnaires to be filled out at home and collected once fulfilled at a later appointment.

The clinical assessment for the whole sample included:Socio-demographic data: gender, age, and self-perceived socio-economic status based on the Hollingshead and Redlich scale [46] were collected by means of a semi-structured interview for parents and adolescents.Past clinical history of medical and mental health problems (patient’s symptoms, illnesses, conditions, developmental problems, and other significant ife events) and mental health diagnosis (based on DSM-V diagnosis) from both adolescents and their family were collected by means of a semi-structured interview.The Columbia-Suicide Severity Rating Scale (C-SSRS) [47] on its Spanish validation by Al-Halabí et al. [48] was administered. The C-SSRS is a semi-structured interview containing six items, including the presence, severity, and frequency of suicidal behavior during the evaluation period, for adolescents.

The adolescent self-reported questionnaires included:
The European Bullying Intervention Project Questionnaire (EBIP-Q) [49] on its Spanish validation by Ortega, del Rey and Casas [50]. The EBIP-Q is a 14-item questionnaire that assesses bullying in high school students: seven items describe the aspects related to victimization and seven items related to aggression, with three dimensions: victim, victimized-aggressor, and aggressor.The European Cyberbullying Intervention Project Questionnaire (ECIP-Q) [51] on its Spanish validation by Ortega, del Rey and Casas [50]. The ECIP-Q is a 22-item questionnaire assessing cyberbullying in high school students with three dimensions: cyber-victimization, cyber-victimized-aggressor and cyber-aggression.The CaMir-R [52] measures attachment and representations of affection and the conception of family functioning in adolescence and early adulthood. The questionnaire consists of 32 items that evaluate three different attachment styles: secure, insecure-ambivalent, and insecure-avoidant.The Parental Bonding Instrument (PBI) [53] by Gómez-Beneyto et al. [54] assesses two components of the parent–child relationship: (1) demonstrations of care (by the parent) and (2) parental overprotection. Optimal bonding is thought to be characterized by high levels of caring and low levels of overprotection. Through its 25 items, we can evaluate four family models: optimal, affectionate-constraint, affectionless-control, and neglectful.The Family Adaptability and Cohesion Evaluation Scale (FACES III) [55] on its Spanish Translation and validation of the FACES p20 version by Martínez-Pampliega, Iraurgi, Galindez, and Sanz [56].It is a 20-item scale that informs about the degree of cohesion and flexibility within the family system perceived by the adolescent within the Olson Circumplex Model framework. The FACES III evaluate 16 types of family functioning specified in three subgroups: compensated family functioning, decompensated family functioning, and very decompensated family functioning.The Stress Life Events Scale (SLES) [57], adolescent version, adapted and validated into Spanish by Rivera y Revuelta and Fumero [58]. The SLES evaluates the stressful life events of the participants through 43 items corresponding to 43 different life events.The Child Behavior Check-List (CBCL) [59]; adapted and validated into Spanish by Sardinero, Pedreira, and Muñiz [60]. The CBCL evaluates emotional and behavioral problems in children and adolescents (6–18 years of age), providing ratings for 20 competence and 120 problem items.

The parents’ self-reported questionnaires included:
The Parental Bonding Instrument (PBI) [53], validated in Spanish by Gómez-Beneyto et al. [54]. The PBI assesses two components of the parent–child relationship: demonstrations of caring (by the parent) and parental overprotection. Optimal bonding is characterized by high levels of caring and low levels of overprotection. Through their 25 items, it is possible to evaluate four family models: optimal, affectionate-constraint, affectionless-control, and neglectful.The Family Adaptability and Cohesion Evaluation Scale (FACES III) [55] validated in a Spanish population by Polaino-Lorente and Martínez-Cano [61]. It is a 40-item scale that informs about the degree of cohesion and flexibility of the family system perceived by the parent within the framework of the Olson Circumplex Model. The FACES III evaluate 16 types of family functioning specified in three subgroups: compensated family functioning, decompensated family functioning, and very decompensated family functioning.The Stress Life Events Scale (SLES) [57] parents’ version adapted and validated into the Spanish version by Rivera y Revuelta and Fumero [58]. The SLES evaluates the stressful life events of the participants through 43 items corresponding to 43 different life events, giving information on the number of traumatic events for parents and adolescents and the interference of the events for parents and adolescents.The Child Behavior Check-List (CBCL) [59] adapted and validated into Spanish by Sardinero, Pedreira, and Muñiz [60]. The CBCL evaluates emotional and behavioral problems in children and adolescents, providing ratings for 20 competence and 120 problem items, giving information for three dimensions: internalizing, externalizing, and total problems.

Suicidal behavior in our study is considered as per Al-Halabí et al. 2021 [62], suggesting that suicidal behavior comprises a set of thoughts and behaviors with suicide intention, and suicide attempts as engaging in potentially self-destructive behavior in which there is at least some intention to die as a result of the behavior. This conceptualization differs from the non-suicidal self-injuries in which the final intention has nothing to do with death.

### 2.3. Procedure

Data collection and coding for the clinical sample occurred between May 2018 and May 2020 and were performed by clinical psychologists and psychiatrists during the patient’s hospital admission to the mental health inpatient service of Hospital Sant Joan de Déu. Data collection and coding for the control sample were carried out in different schools and recreational institutions during the same period. Clinical psychologists and psychiatrists assessed both samples in a 2-h session with the adolescents and their parents. All participants (parents and adolescents) gave their written consent after receiving the information regarding the study, its objectives, and the agreement of confidentiality and protection of personal data. Participation in the study was not remunerated. 

### 2.4. Ethical Aspects

The study complies with the internal regulations of the Hospital Ethics and Research Committee of Hospital Sant Joan de Déu and has its approval (with the internal code PIC-158-18) as well as that of the World Medical Association and the Declaration of Helsinki of 1995 [63] with its successive amendments. Since no additional measures were collected or any other invasive procedures were performed on patients, no additional informed consents were required rather than the standard one provided during the hospital admission stay in the hospital or their school/recreational affiliations. All participants (parents and adolescents) gave their written consent after receiving the information regarding the study, its objectives, and the agreement of confidentiality and protection of personal data. Participation in the study was not remunerated. 

### 2.5. Data Analyses

The data were analyzed with the IBM SPSS statistics version 25. Descriptive statistics of all variables were performed. Paired controls by sex and age were performed between case and control samples. Descriptive statistics and frequency distribution analyses of all variables considered in the present study were calculated. Differences between the groups were analyzed with the Student’s t-test and one-way variance analysis (ANOVA) for independent samples (for quantitative variables) or a chi-squared test (or the Fisher’s exact test, when no application criteria were met for the chi-squared), calculated from 2 × 2 contingency tables (for categorical variables). Cohen’s d was calculated to analyze size effects. The significance of all tests was set at a probability level of 5% or less, with a 95% confidence interval and a high effect size (d greater than or equal to 0.8, Eta value where appropriate), always indicating the exact value offered by the statistical package SPSS.

## 3. Results

### 3.1. Sample Description

Data from 100 participants (74% females *n* = 74; age *M* ± *S.d.* = 15.01 ± 1.54) were collected during two years. The sample consists of 50 controls (females: *n* = 34, 68%; age *M* ± *S.d.* = 14.80 ± 1.73) and 50 cases (females: *n* = 40, 80%; age *M* ± *S.d.* = 15.22 ± 1.31). 

There is no significant difference in age (*t_7.497_ = 1.369*, *p = 0.174*, *CI 95% −0.189–1.029*) (cases = 15.22 ± 1.31 vs. controls 14.80 ± 1.73) or gender between groups, despite a higher percentage of females (80%) in the case group (✗*^2^* = 1.871, *df* = 1, *p* = 0.171, *Eta* = 0.137). 

Other clinical characteristics of the studied sample are summarized in Table 1.

Concerning academic performance, adolescents from the case group tend to repeat, more frequently, an academic year (✗*^2^* = 5.263, *df* = 1, *p* = 0.022, *Eta* = 0.229). In this sense, the number of repeated academic years differs significantly between groups (*t_27.004_ = 2.474*, *p = 0.016*, *CI 95% 0.043–0.397*), with cases repeating more times than controls a mean of 0.32 *±* 0.55 academic years and controls 0.1 *±* 0.30.

### 3.2. Family Relationship Differences between Cases and Controls

Table 2 Displays bonding and attachment characteristics of the studied sample according to each group (cases vs. controls).

Concerning attachment styles (CaMir), significant differences were observed between groups, with a higher prevalence of insecure-avoidant attachment style in cases (58%) compared to controls (✗*^2^* = 14.760, *df* = 2, *p* = 0.001, *Eta* = 0.384). Security revealed to be a protective factor for suicide behavior (*p* < 0.001, *Cohen’s d* = 1.04 large size effect), while the following variables were identified as risk factors: Family concern (*p* = 0.001, *Cohen’s d* = 0.67 medium-size effect), Parental interference (*p* = 0.002, *Cohen’s d = 0*.65 medium-size effect), Self-sufficiency and resentment towards parents (*p* < 0.001, *Cohen’s d* = 2.38 large size effect), and Childhood trauma (*p* = 0.011, *Cohen’s d* = 0.47 medium-size effect). There were not significant differences between groups concerning authority or parental permittivity dimensions of the CamiR. 

Parental bonding (PBI) provides different profiles with significant differences between adolescents and parents in the clinical group. When analyzing the adolescent self-reported results, affectionless-control parents (42%) were associated with suicidal behavior (✗*^2^* = 21.940, *df* = 3, *p* < 0.001, *Eta* = 0.468), therefore, appearing as a risk factor. Similarly, when analyzing the parents’ responses to the PBI, negligent parents (38%) appeared also as risk factors since this profile was significantly related to suicidal behavior in adolescents (✗*^2^* = 22.054, *df* = 3, *p* < 0.001, *Eta* = 0.470). On the other hand, and as expected, both parents’ and adolescents’ responses to the PBI coincided to point out optimal parenting as a clear protective factor for suicidal behavior. Other protective factors were care assessed by parents (*p* < 0.001, *Cohen’s d =* 0.94 large size effect) and care assessed by adolescents (*p* < 0.001, *Cohen’s d =* 1.26 large size effect). Additionally, overprotection assessed by parents (*p* < 0.001, *Cohen’s d =* 0.94 large size effect) became a protection factor for suicide behaviors. There were no significant differences between cases and controls in overprotection observed by adolescents from the PBI. 

Finally, results from the FACES III showed that compensated family functioning (54%) is considered a protective factor for suicidal behavior (✗*^2^* = 7.162, *df* = 2, *p* = 0.028, *Eta* = 0.268). There were not significant differences in cohesion and adaptability between cases and controls evaluated by parents with the FACES III. The functioning assessed by the FACESp20 shows Cohesion in the family as a protective factor (*p* < 0.001). 

### 3.3. Comparison between Cases and Controls of the Traumatic Events Variables of the Assessment

Table 3 Displays the significant differences concerning traumatic life events between groups (cases vs. controls).

Bullying (EBIP-Q) is considered a risk factor for suicidal behavior. In our study, we found that victimization-bullying (*p* < 0.001, *Cohen’s d =* 1.45 large size effect) and aggression-bullying (*p* < 0.001, *Cohen’s d =* 1.04 large size effect) were risk factors for suicide behaviors in the comparison between cases and controls.

Similarly, being a cybervictim (ECIP-Q) appeared to be a suicidal behavior risk factor, with significant differences between cases and controls in victimization-cyberbullying (*p* < 0.001, *Cohen’s d =* 0.98 large size effect) and aggression-cyberbullying (*p* = 0.002, *Cohen’s d =* 0.66 medium-size effect). 

Other risk factors for suicide behavior were the total number of stressful live events (SLEs) reported by the adolescents (*p* < 0.001, *Cohen’s d* = 1.73 large size effect) and their parents (*p* < 0.001, *Cohen’s d =* 1.20 large size effect), and the interference of the SLEs responded by adolescents (*p* < 0.001, *Cohen’s d =* 2.12 large size effect), and by their parents (*p* < 0.001, *Cohen’s d =* 1.54 large size effect).

### 3.4. Comparison between Cases and Controls of the Mental Health Problems Variables of the Assessment

Table 4 shows the significant differences in the CBCL between groups. As it can be observed, Internalizing problems (*p* < 0.001, *Cohen’s d =* 4.25 large size effect), Externalizing problems (*p* < 0.001, *Cohen’s d =* 3.49 large size effect), and Total problems (*p* < 0.001, *Cohen’s d =* 3.97 large size effect) were higher in the cases group.

## 4. Discussion

The general aim of this research was to analyze the association between family relationship variables and the occurrence of stressful life events to suicide behavior in adolescents.

Overall, the clinical sample of the present study was similar to other studies in terms of age and gender, showing a higher percentage of females and mean age around 14–15 years old [64,65]. Concerning suicidal behavior, it was observed that 36% of the studied sample repeated the attempt, revealing higher rates compared to similar studies in this field with this specific population sample, which is around 18–20% [45,66,67]. These differences could be explained due to different inclusion/exclusion criteria and the recruitment strategy. In this sense, the present research invited adolescents from an inpatient unit in a tertiary hospital, during the first hours after being hospitalized due to the risk of repetition or the medical consequences after the first attempt. Therefore, it can be inferred that this is a very vulnerable population with very specific clinical characteristics. In our sample, academic performance was also related to suicidal behavior as in other studies [68,69], both showing school failure by repeating a course, and the fact that the greater the number of repeated courses, the greater the risk of repeating the suicide attempt. This could be explained by the stress that might involve repeating a course that could cause a certain sense of failure, low self-esteem and could also trigger new stressful situations like meeting new classmates or being older than the rest of the new classmates [70]. In this sense, the study of McBee-Strayer and colleagues explained that old-for-grade students were also more likely to report a suicide attempt with increased suicidal ideation and planning, suggesting that common risk factors for suicide repetition seem to be anxiety, substance use, depression symptoms, and others [71].

When focusing on attachment and family-related variables explored in this research, a significant difference with the insecure-avoidant attachment style was found, with higher prevalence observed in the clinical sample of adolescents, in line with previous research [72].This is evidence of the needed confidence between parents and adolescents shown in other studies of the literature [73,74]. Similarly, secure attachment (assessed by means of the CamiR) was revealed as a protective factor for suicide. This was also found in the study by McLaughlin and colleagues, which showed that a greater secure attachment predicted lower rates of internalizing disorders in both genders [75], from which we can infer that this would reduce suicidal behavior, as internalizing disorders are a risk factor for suicidal behavior [76,77]. Additionally, childhood trauma appears repeatedly as a risk factor in different studies of suicide behaviors in adolescents [78,79] and can be related to the stressful life events as observed in the present research.

In regard with the parental bonding variables, a relationship between negligent and affection-less parenting and suicidal behavior was found, as previously shown in the literature [15,17]. In this sense, the variable care expressed by parents and adolescents, and the overprotection assessed by parents, are considered as a protective factor for suicide. Previous studies have found that low family care [80], and living in dysfunctional households [81] are risk factors for suicide behaviors, while an increased ability to care as an element that could reduce the number of suicides behaviors [82].

When focusing on family functioning variables and their association with suicidal behavior, a positive relationship between compensated family functioning (as in the Circumplex Model) and controls was found. Previous scientific literature has already pointed out that having a decompensated family functioning is a risk factor for suicidal behavior [83]. Some research has also found that families with adaptability problems are at higher risk of suicidal behaviors [83]. On the contrary, higher cohesion in the family is a protective factor for depressive symptoms in adolescents [84] and this has served as a basis for research studying family cohesion as a protective factor for suicidal behavior in young people [85]. However, more studies are still required in this field to deepen these associations.

When focusing on stressful life events, a significant relationship between past trauma events and suicidal behavior, and between bullying and cyberbullying with suicidal behaviors, was found in the present research. This is in line with most research in this field, pointing out this clear relationship between different forms of bullying and suicidal behavior [86].In our sample, there is a significant relationship between the group of cases and the victimized aggressors, which is in line with previous studies that portray the victims-aggressive young people as being more maladjusted than their peers in terms of their social and emotional functioning [87]. Additionally, concerning cyberbullying, a significant relationship between cyber victims and cases was identified, as stressed out in previous studies, which strongly relates them to suicidal behaviors [88,89]. We can also find similar results regarding bullying and cyberbullying in the quantitative items, having positive results for victimization and aggression as risk factors for suicidal behavior in adolescents, as shown in previous studies, like the one by Hinduja and Patchin in 2010 in which both factors, aggression and victimization, are more likely to trigger suicidal thoughts and suicide attempts [90]. On the other hand, the accumulation of different stressful life events (not only bullying in its different forms) has also been revealed as a risk predictor of suicidal behavior (the higher the number of stressful life events, the higher the risk of having suicide behaviors), and also it happens with the affectation caused by this stressful life events (the higher affectation, the higher risk of having suicide behaviors). This has been reported in previous theoretical and research studies about suicide in adolescents. For example, Yildiz reported that stressful life events increase suicidality in adolescents, partly by increasing psychological distress and eroding perceived social support, giving some valuable clues to some possible preventive strategies [43].

Finally, our study found a significant relationship between mental health problems (internalizing, externalizing, and total symptoms) and suicide behavior. Specifically, several diagnoses have been associated with suicidal behavior. Depressive symptoms are most common, but also anxiety, affective disorders, disruptive behavior, and substance disorders were important variables for suicide behaviors in adolescents [91].

It is important to note that this research is not exempt from limitations. First, the clinical sample (cases) was recruited from a unique center and, therefore, results cannot be generalized to other clinical samples from different settings and/or even countries or outside our influence area. However, it is true that the hospital from which the sample was recruited is a reference center in our country for mental health problems in children and adolescents. It is also important to note that the evaluation of cases and their parents is done during the inpatient hospital stay, so the situation can generate a bias in the answers, although all the evaluations were administered when the patient and the family had overcome the first moment of acute crisis, being the time of evaluation closer to discharge than to admission, once the intervention, and the patient’s own evolution, allowed it. In addition, being a study with a relatively small sample, there is focus on a proper description and characterization of the two samples, rather than conducting complex statistical analyses, believing that the strong point of this research is to compare not only a sample of adolescents with their peers, but also that of their parents. Despite this, studies with a larger sample size are required to be able to carry out more complex predictive statistical models, and continue to provide knowledge in this relevant field of research.

We also believe this study has some strength. This is a case-control study and the assessment of suicidal behavior in adolescents in our area has been scarce so far. It is also important to note that family factors in suicidal behavior in adolescents have been little studied.

Considering main results, it can be concluded that insecure attachment and rigid or negligent relationships between parents and their offspring, bullying and cyberbullying, and stressful life events are clear and significant risk factors for suicidal behavior in adolescents, whereas having a good family functioning with care, security, and flexibility have been revealed as key protective factors. In regards to stressful life events, we think this is a serious matter as we not only observe higher rates of stressful life events in cases but also family relational patterns with less capacity for containment, such as a higher prevalence of an insecure-avoidant attachment style, in addition to unbalanced family functions, with the parent–child relationship being more neglectful and less affectionate. These situations make adolescents even more vulnerable both to their peers and the environment. This, in addition to lack of family support, seems to be sufficient elements that can explain the poor prognosis of these cases, with such high levels of relapse. 

Clinical implications of this research are multiple. First of all, the importance of helping improve socio-emotional skills becomes evident in assisting adolescents in facing bullying and cyberbullying situations, which unfortunately have appeared quite common in this age range, and this is applicable not only from the victims but also from their peers. In the same way, there is a clear need to incorporate a family approach to the treatment of adolescents with suicidal behavior, since our research has found out that there are relationship models that might play a protective function. Family-based therapies have great potential to prevent suicidal behaviors in adolescents [92].For this reason, more research is needed to address the relevance of family interventions in this population, improving knowledge about family-related risk and protective factors, and facilitating family treatments to address suicidal behaviors.

## Figures and Tables

**Table 1 ijerph-19-09892-t001:** Clinical characteristics of the sample (*N* = 100).

		Cases (%) (*n*)	Controls (%) (*n*)
Suicidal behavior	Suicide thoughts	34% (17)	0 (0)
	Self-destructive behavior	8% (4)	0 (0)
	Suicide attempt	54% (27)	0 (0)
	Previous suicidal behavior	36% (18)	0 (0)
Family demography	Single parents	16% (8)	12% (6)
	Original family with both parents	48% (24)	50% (25)
	Divorced parents living with both	12% (6)	26% (13)
	Other types of family	24% (12)	12% (6)
Family studies	Low	32% (16)	32% (16)
	Medium	46% (23)	24% (12)
	High	22% (11)	44% (22)
Professional situation	Housewives/husbands	8% (4)	0 (0)
	Active workers	80% (40)	84% (42)
	Retired	0 (0)	8% (4)
	Unemployed	12% (6)	8% (4)
Skilled employment done	Low	34% (17)	22% (11)
	Medium	34% (17)	28% (14)
	High	32% (16)	50% (25)
Repeated course	Yes	28% (14)	10% (5)
	No	72% (36)	90% (45)
Number of repeated courses	0	72% (36)	90% (45)
	1	24% (12)	10% (5)
	2	4% (2)	0 (0)
Clinical data	Previous mental health diagnosis	100% (50)	0 (0)
	Comorbid diagnosis	36% (18)	0 (0)
	Family background of mental health diagnosis	66% (33)	0 (0)
	Family history of suicide behaviors	16% (8)	0 (0)

**Table 2 ijerph-19-09892-t002:** Comparison of the family relationship variables (attachment, parental bonding, and family functioning) between cases and controls (*n* cases = 50, *n* controls = 50).

Test	Sub-Sample	Median	*S.d.*	*S.e.*	*Cohen’s d*	*t*	df	*p* Value (Bilateral)	95% Confidence Interval of the Difference	
									Inferior	Superior
PBI parents care	Case	26.34	6.73	0.95	0.94	−4.69	98	<0.001	−7.57	−3.07
	Control	31.66	4.36	0.62						
PBI overprotection parents	Case	11.34	4.79	0.67	0.94	−4.69	98	<0.001	−7.57	−3.07
	Control	9.70	3.05	0.43						
PBI adolescent care	Case	21.40	8.36	1.18	1.26	−6.28	87.68	<0.001	−11.93	−6.19
	Control	30.46	5.85	0.83						
CamiR Security	Case	33–90	18.00	2.55	1.04	−5.20	98	<0.001	−20.80	−9.31
	Control	48.96	9.75	1.38						
CamiR Family Concern	Case	53.35	11.78	1.67	0.67	3.33	98	0.001	2.64	10.45
	Control	46.80	7.39	1–05						
CamiR Parental interference	Case	58.07	12.80	1.81	0.65	3.27	82.07	0.002	2.73	11.21
	Control	51.10	7.97	1.13						
CamiR Self-sufficiency and resentment towards parents	Case	66.51	9.05	1.28	2.38	11.92	98	<0.001	17.59	24.62
	Control	45.40	8.65	1.22						
Camir Childhood trauma	Case	84.08	90.84	12.85	0.47	2.66	50.27	0.011	8.38	60.32
	Control	49.73	10.34	1.46						
Cohesion FACESp20	Case	27.82	9.70	1.37	1.11	−5.57	98	<0.001	−13.24	−6.28
	Control	37.58	7.71	1.09						

*S.d.:* Standard deviation. *S.e.:* Standard error mean.

**Table 3 ijerph-19-09892-t003:** Comparison of traumatic events (SLES, Bullying, and Cyberbullying) between cases and controls (*n* cases = 50, *n* controls = 50).

Test	Case Control	Median	*S.d.*	*S.e.*	*Cohen’s d*	*t*	*df*	*p* Value (Bilateral)	95% Confidence Interval of the Difference	
									Inferior	Superior
Number of events SLES adolescents	Case	28.78	18.03	2.55	1.73	8.63	56.61	<0.001	17.54	28.14
	Control	5.94	5.04	0.71						
Interference SLES adolescents	Case	68.46	38.21	5.40	2.12	10.60	54.23	<0.001	47.63	69.85
	Control	9.72	8.76	1.24						
Number of events SLES parents	Case	21.44	18.35	2.60	1.20	6.01	54.86	<0.001	10.70	21.42
	Control	5.38	4.49	0.64						
Interference SLES parents	Case	45.42	31.94	4.52	1.54	7.68	56.21	<0.001	26.58	45.34
	Control	9.46	8.69	1.23						
EBIP-Q Victimization Bullying	Case	8.44	7.51	1.06	1.45	7.25	51.61	<0.001	5.64	9.96
	Control	0.64	1.23	0.17						
EBIP-Q Aggression Bullying	Case	2.36	2.99	0.42	1.04	5.22	50.35	<0.001	1.37	3.08
	Control	0.14	0.35	0.05						
EBIP-Q Victimization Cyberbullying	Case	4.76	5.92	0.84	0.98	4.91	51.17	<0.001	2.06	7.36
	Control	0.60	0.88	0.13						
EBIP-Q Aggression Cyberbullying	Case	1.94	3.35	0.47	0.66	3.29	51.40	0.002	0.62	2.54
	Control	0.36	0.53	0.07						

*S.d.:* Standard deviation. *S.e.:* Standard error mean.

**Table 4 ijerph-19-09892-t004:** Comparison of mental health problems (CBCL) between cases and controls (*n* cases = 50, *n* controls = 50).

Test	Case Control	Median	*S.d.*	*S.e.*	*Cohen’s d*	*t*	*df*	Sig. (Bilateral)	95% Confidence Interval of the Difference	
									Inferior	Superior
CBCL Internalizing	Case	85.12	28.21	3.99	4.25	21.24	98	<0.001	76.86	92.70
	Control	0.34	0.82	0.12						
CBCL Externalizing	Case	77.34	31.25	4.42	3.49	17.44	98	<0.001	68.31	85.85
	Control	0.26	0.85	0.12						
CBCL Total	Case	77.70	27.05	3.83	3.97	19.85	98	<0.001	68.58	83.82
	Control	1.5	2.24	0.32						

*S.d.:* Standard deviation. *S.e*.: Standard error mean.

## Data Availability

The data that support the findings of this study are available from the corresponding author upon reasonable request.

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
