# Peer review of "Family Factors Related to Suicidal Behavior in Adolescents"

_ijerph, 2022, doi:10.3390/ijerph19169892_

Round 1
Reviewer 1 Report
Thank you for the opportunity to review this paper. It will have significance for clinicians and therapists who provide services to young adults and adolescents and it adds to the body of knowledge on suicide and suicidal behaviour in adolescents. There are a few items that I need to be addressed to enhance this paper for publication.
Methods: The authors maintain that they conducted semi-structured interviews. However, there is not much evidence of what was asked of participants in these interviews, were they used to administer all the questionnaires or were other questions asked of participants? Were the adolescents interviewed separately to the parents or were they interviewed together? Perhaps this could be expanded upon.
Results: In describing the participants mention is made that 74% were female the n (number) should also be reported alongside the percentages. Table 1: Should also include the n (number) as well as the percentages as not everyone may have answered all these questions. On page 9 live350 the authors maintain that Aggression-bullying (p<0.001, Cohen's d=1.45 large size effect) however the Table 3 above states 1.04 large size effect? This needs to be corrected.
Author Response
Thank you for your reviews. We answer your questions and recommendations in doc file attached.

Reviewer 2 Report
This is an interesting manuscript detailing a study conducted with a high-risk group for self-harm and suicidality in a clinical setting, focusing on a number of key risk factors for suicidality amongst younger adolescents in contact with mental health services.
I do have some concerns about the novelty of this study (in terms of the contribution this study makes to the existing literature on attachment and family environments and their role in suicidality) and in the choices made by the authors in terms of the data analyses presented.
General Comments:
Aims/hypotheses: there seemed to be a lack of clear hypotheses at the end of the introduction and some mismatch between the the stated aims at the end of the intro with the beginning of the discussion section. The authors seem to suggest that they are looking at the relationship between these different factors (attachment, stressful life events, suicidal behaviours) but the presented results do not really investigate any clear associations between these different factors (e.g. inter-relationships/interactions between attachment*family environment). The results presented really only show some basic differences in means between groups, not a clear sense of how these factors interact - the latter may be an important point for clinical interventions.
Analysis: in relation to the above, it's not clear why the authors did not conduct logistic regressions or equivalent analyses to explore how these factors together predict those at high/low risk for suicide. The authors present a large amount of tests without adjusting for multiple comparisons (i.e. an adjusted alpha to account for inflated error rates with multiple comparisons - Bonferroni or similar). Because of these multiple comparisons, it is difficult to understand which factors are important to consider. As mentioned above, given the overlapping/interacting nature of the variables studies, why didn't the authors explore such interacting relationships given their clinical utility?
Introduction/Discussion: I found these sections to generally be quite long and overly worded - it would be useful for these sections to be much more concise (e.g. cutting out the descriptive background detail on attachment/the Strange Situation).
The attachent section (starting Line 53) is written in a different tone to the previous sections - please check that this is not plagiarised from other sources and ensure that the written expression is consistent with other sections.
In various places, the text size/font appears to change, please check that the formatting is consistent throughout the manuscript.
Referencing: should be using the journal's numbered format as outlined in the author instructions.
Specific Comments:
Line 40 - 'significant' or 'large' would be a better word than 'big'
Line 48 - please give an example of what 'dysfunctional family patterns' are being referred to here.
Line 84 - I found the latter part of this sentence referring to 'mediation by others' to be unclear in terms of wording/what is being referred to. Please edit/clarify the point here, especially which aspects of personality are important to consider.
Line 101 - this paragraph read to me like a more opening paragraph/earlier paragraph in the introduction - I was puzzled why this appeared after the section on attachment.
Line 143 - no clear hypotheses are presented here. Given the existing literature, the authors should be able to identify potential hypotheses relationships between the study variables.
Line 154 - there is mention of participants being recruited based on suicidal behaviour but the bracketed information refers to ideation/planning which are not behavioural actions. Were the participants referred to the sampled service based on their suicide risk rather than a behaviour?
Line 156 - what are 'ludic associations'?
Line 169 - perhaps delete 'both included'? Wasn't that clear what was being referred to here.
Section 2.4 (from Line 267) - please clarify in this section that ethics approval was obtained for the study (only mentions study was conducted in line with ethics guidance).
Line 288 - how do you set 'a high effect size'?
Table 2 (but check all tables) - some of the text in the main body of the text is boldened, I think this is a mistake in the formatting?
Line 428 - this sentence is difficult to read, it seems like there are some missing punctuation here and ambiguous writing.
Author Response

(The authors gave the same response as above.)

Round 2
Reviewer 2 Report
Thank you to the authors for comprehensively addressing my comments on the previous version of the manuscript. I feel that the manuscript has been substantially improved in clarity and reporting.
I have one further comment/request - please could the authors add a point to the Discussion's limitations section to acknowledge their points about the more descriptive nature of their analyses. It would be appropriate to highlight that multiple comparisons had been made here and that more complex/nuanced analyses would be required to better elucidate which factors are important in adolescent suicidality as per their study (i.e. future studies would benefit from investigating multivariate analyses of these factors over time, such as controlling for various confounders/covariates in logistic regression models). Something based on their response to my Point 2 in their response letter to my original comments would be appropriate in the manuscript to acknowledge the limitations of the analyses conducted in the study.
Author Response
Thanks again for your review.
